# A Walsh Hadamard Derived Linear Vector Symbolic Architecture

**Mohammad Mahmudul Alam**[1], **Alexander Oberle**[1], **Edward Raff**[1,2], **Stella Biderman**[2],
**Tim Oates**[1], **James Holt**[3]
[1]University of Maryland, Baltimore County,[2]Booz Allen Hamilton,
[3] Laboratory for Physical Sciences
`m256@umbc.edu, aoberle1@umbc.edu, Raff_Edward@bah.com,`
`biderman_stella@bah.com, oates@cs.umbc.edu, holt@lps.umd.edu`

## Abstract

Vector Symbolic Architectures (VSAs) are one approach to developing Neuro-symbolic AI, where two vectors in $\mathbb{R}^d$ are 'bound' together to produce a new vector in the same space. VSAs support the commutativity and associativity of this binding operation, along with an inverse operation, allowing one to construct symbolic-style manipulations over real-valued vectors. Most VSAs were developed before deep learning and automatic differentiation became popular and instead focused on efficacy in hand-designed systems. In this work, we introduce the Hadamard-derived linear Binding (HLB), which is designed to have favorable computational efficiency, and efficacy in classic VSA tasks, and perform well in differentiable systems. Code is available at `https://github.com/FutureComputing4AI/Hadamard-derived-Linear-Binding`.

## 1 Introduction

Vector Symbolic Architectures (VSAs) are a unique approach to performing symbolic style AI. Such methods use a binding operation $\mathcal{B} : \mathbb{R}^d \times \mathbb{R}^d \longrightarrow \mathbb{R}^d$, where $\mathcal{B}(x, y) = z$ denotes that two concepts/vectors $x$ and $y$ are connected to each other. In VSA, any arbitrary concept is assigned to vectors in $\mathbb{R}^d$ (usually randomly). For example, the sentence "the fat cat and happy dog" would be represented as $\mathcal{B}(fat, cat) + \mathcal{B}(happy, dog) = S$. One can then ask, "what was happy" by *unbinding* the vector for happy, which will return a noisy version of the vector bound to happy. The unbinding operation is denoted $\mathcal{B}^*(x, y)$, and so applying $\mathcal{B}^*(S, happy) \approx dog$.

Because VSAs are applied over vectors, they offer an attractive platform for neuro-symbolic methods by having natural symbolic AI-style manipulations via differentiable operations. However, current VSA methods have largely been derived for classical AI tasks or cognitive science-inspired work. Many such VSAs have shown issues in numerical stability, computational complexity, or otherwise lower-than-desired performance in the context of a differentiable system.

As noted in [39], most VSAs can be viewed as a linear operation where $\mathcal{B}(a, b) = a^\top G b$ and $\mathcal{B}^*(a, b) = a^\top F b$, where $G$ and $F$ are $d \times d$ matrices. Hypothetically, these matrices could be learned via gradient descent, but would not necessarily maintain the neuro-symbolic properties of VSAs without additional constraints. Still, the framework is useful as all popular VSAs we are aware fit within this framework. By choosing $G$ and $F$ with specified structure, we can change the computational complexity from $\mathcal{O}(d^2)$, down to $\mathcal{O}(d)$ for a diagonal matrix.

In this work, we derive a new VSA that has multiple desirable properties for both classical VSA tasks, and in deep-learning applications. Our method will have only $\mathcal{O}(d)$ complexity for the binding step, is numerically stable, and equals or improves upon previous VSAs on multiple recent deep

38th Conference on Neural Information Processing Systems (NeurIPS 2024).

learning applications. Our new VSA is derived from the Walsh Hadamard transform, and so we term our method the Hadamard-derived linear Binding (HLB) as it will avoid the $\mathcal{O}(d \log d)$ normally associated with the Hadamard transform, and has better performance than more expensive VSA alternatives.

Related work to our own will be reviewed in section 2, including our baseline VSAs and their definitions. Our new HLB will be derived in section 3, showing it theoretically desirable properties. section 4 will empirically evaluate HLB in classical VSA benchmark tasks, and in two recent deep learning tasks, showing improved performance in each scenario. We then conclude in section 5.

## 2 Related Work

Smolensky [38] started the VSA approach with the Tensor Product Representation (TPR), where $d$ dimensional vectors (each representing some concept) were bound by computing an outer product. Showing distributivity ($\mathcal{B}(\boldsymbol{x}, \boldsymbol{y} + \boldsymbol{z}) = \mathcal{B}(\boldsymbol{x}, \boldsymbol{y}) + \mathcal{B}(\boldsymbol{x}, \boldsymbol{z})$) and associativity, this allowed specifying logical statements/structures [13]. However, for $\rho$ total items to be bound together, it was impractical due to the $\mathcal{O}(d^\rho)$ complexity. [36, 25, 24] have surveyed many of the VSAs available today, but our work will focus on three specific alternatives, as outlined in Table 1. The Vector-Derived Transformation Binding (VTB) will be a primary comparison because it is one of the most recently developed VSAs, which has shown improvements in what we will call "classic" tasks, where the VSA's symbolic like properties are used to manually construct a series of binding/unbinding operations that accomplish a desired task. Note, that the VTB is unique in it is non-symmetric ($\mathcal{B}(\boldsymbol{x}, \boldsymbol{y}) \neq \mathcal{B}(\boldsymbol{y}, \boldsymbol{x})$). Ours, and most others, are symmetric.

Table 1: The binding and initialization mechanisms for our new HLB with baseline methods. HLB is related to the HRR in being derived via a similar approach, but replacing the Fourier transform $\mathcal{F}(\cdot)$ with the Hadamard transform (which simplifies out). The MAP is most similar to our approach in mechanics, but the difference in derived unbinding steps leads to dramatically different performance. The VTB is the most recently developed VSA in modern use. The matrix $V_{\boldsymbol{y}}$ of VTB is a block-diagonal matrix composed from the values of the $\boldsymbol{y}$ vector, which we refer the reader to [12] for details. The TorchHD library [15] is used for implementations of prior methods.

| METHOD | BIND $\mathcal{B}(x,y)$ | UNBIND $\mathcal{B}^*(x,y)$ | INIT $x$ |
|---|---|---|---|
| HRR | $\mathcal{F}^{-1}(\mathcal{F}(\boldsymbol{x}) \odot \mathcal{F}(\boldsymbol{y}))$ | $\mathcal{F}^{-1}(\mathcal{F}(\boldsymbol{x}) \oslash \mathcal{F}(\boldsymbol{y}))$ | $x_i \sim \mathcal{N}(0, 1/d)$ |
| VTB | $V_y x$ | $V_y^\top x$ | $\tilde{x}_i \sim \mathcal{N}(0,1) \rightarrow x = \tilde{\boldsymbol{x}}/\|\tilde{\boldsymbol{x}}\|_2$ |
| MAP-C | $x \odot y$ | $x \odot y$ | $x_i \sim \mathcal{U}(-1, 1)$ |
| MAP-B | $x \odot y$ | $x \odot y$ | $x_i \sim \{-1, 1\}$ |
| HLB | $x \odot y$ | $x \oslash y$ | $x_u \sim \{\mathcal{N}(-\mu, 1/d), \mathcal{N}(\mu, 1/d)\}$ |

Next is the Holographic Reduced Representation (HRR) [32], which can be defined via the Fourier transform $\mathcal{F}(\cdot)$. One derives the inverse operation of the HRR by defining the one vector $\vec{1}$ as the identity vector and then solving $\mathcal{F}(\boldsymbol{a}^*)_i \mathcal{F}(\boldsymbol{a})_i = 1$. We will use a similar approach to deriving HLB but replacing the Fourier Transform with the Hadamard transform, making the HRR a key baseline. Last, the Multiply Add Permute (MAP) [10] is derived by taking only the diagonal of the tensor product from [38]'s TPR. This results in a surprisingly simple representation of using element-wise multiplication for both binding/unbinding, making it a key baseline. The MAP binding is also notable for its continuous (MAP-C) and binary (MAP-B) forms, which will help elucidate the importance of the difference in our unbinding step compared to the initialization avoiding values near zero. HLB differs in devising for the unbinding step, and we will later show an additional corrective term that HLB employs for $\rho$ different items bound together, that dramatically improve performance.

Our motivation for using the Hadamard Transform comes from its parallels to the Fourier Transform (FT) used to derive the HRR and the HRR's relatively high performance. The Hadamard matrix has a simple recursive structure, making analysis tractable, and its transpose is its own inverse, which simplifies the design of the inverse function $\mathcal{B}^*$. Like the FT, WHT can be computed in log-linear time, though in our case, the derivation results in linear complexity as an added benefit. The WHT is already associative and distributive, making less work to obtain the desired properties. Finally, the WHT involves only $\{-1, 1\}$ values, avoiding numerical instability that can occur with the HRR/FT.

This work shows that these motivations are well founded, as they result in a binding with comparable or improved performance in our testing.

Our interest in VSAs comes from their utility in both classical symbolic tasks and as useful priors in designing deep learning systems. In classic tasks VSAs are popular for designing power-efficient systems from a finite set of operations [14, 23, 17, 30]. HRRs, in particular, have shown biologically plausible models of human cognition [21, 5, 40, 6] and solving cognitive science tasks [8]. In deep learning the TPR has inspired many prior works in natural language processing [34, 16, 35]. To wit, The HRR operation has seen the most use in differentiable systems [43, 41, 42, 27, 29, 33, 1, 2, 28]. To study our method, we select two recent works that make heavy use of the neuro-symbolic capabilities of HRRs. First, an Extreme Multi-Label (XML) task that uses HRRs to represent an output space of tens to hundreds of thousands of classes $C$ in a smaller dimension $d < C$ [9], and an information privacy task that uses the HRR binding as a kind of "encrypt/decrypt" mechanism for heuristic security [3]. We will explain these methods in more detail in the experimental section.

## 3 Methodology

First, we will briefly review the definition of the Hadamard matrix $H$ and its important properties that make it a strong candidate from which to derive a VSA. With these properties established, we will begin by deriving a VSA we term HLB where binding and unbinding are the same operation in the same manner as which the original HRR can be derived [32]. Any VSA must introduce noise when vectors are bound together, and we will derive the form of the noise term as $\eta^\circ$. Unsatisfied with the magnitude of this term, we then define a projection step for the Hadamard matrix in a similar spirit to '[9]'s complex-unit magnitude projection to support the HRR and derive an improved operation with a new and smaller noise term $\eta^\pi$. This will give us the HLB bind/unbind steps as noted in Table 1.

Hadamard $H_d$ is a square matrix of size $d \times d$ of orthogonal rows consisting of only $+1$ and $-1$s given in Equation 1 where $d = 2^n \ \forall \ n \in \mathbb{N} : n \geqslant 0$. Bearing in mind that Hadamard or Walsh-Hadamard Transformation (WHT) can be equivalent to discrete multi-dimensional Fourier Transform (FT) when applied to a $d$ dimensional vector [26], it has additional advantages over Discrete Fourier Transform (DFT). Unlike DFT, which operates on complex $\mathbb{C}$ numbers and requires irrational multiplications, WHT only performs calculations on real $\mathbb{R}$ numbers with addition and subtraction operators and does not require any irrational multiplication.

$$H_1 = [1] \qquad H_2 = \begin{bmatrix} 1 & 1 \\ 1 & -1 \end{bmatrix} \qquad \cdots \qquad H_{2^n} = \begin{bmatrix} H_{2^{n-1}} & H_{2^{n-1}} \\ H_{2^{n-1}} & -H_{2^{n-1}} \end{bmatrix} \qquad (1)$$

Vector symbolic architectures (VSA), for instance, Holographic Reduced Representations (HRR) employs circular convolution to represent compositional structure which is computed using Fast Fourier Transform (FFT) [32]. However, it can be numerically unstable due to irrational multiplications of complex numbers. Prior work [9] devised a projection step to mitigate the numerical instability of the FFT and it's inverse, but we instead ask if re-deriving the binding/unbinding operations may yield favorable results if we use the favorable properties of the Hadamard transform as given in Lemma 3.1.

**Lemma 3.1** (Hadamard Properties). *Let $H$ be the Hadamard matrix of size $d \times d$ that holds the following properties for $x, y \in \mathbb{R}^d$. First, $H(Hx) = dx$, and second $H(x + y) = Hx + Hy$.*

The bound composition of two vectors into a single vector space is referred to as BINDING. The knowledge retrieval from a bound representation is known as UNBINDING. We define the binding function by replacing the Fourier transform in circular convolution with the Hadamard transform given in Definition 3.1. We will denote the binding function four our specific method by $\mathcal{B}$ and the unbinding function by $\mathcal{B}^*$.

**Definition 3.1** (Binding and Unbinding). The binding of vectors $x, y \in \mathbb{R}^d$ in Hadamard domain is defined in Equation 2 where $\odot$ is the elementwise multiplication. The unbinding function is defined in a similar fashion, i.e., $\mathcal{B} = \mathcal{B}^*$. In the context of binding, $\mathcal{B}(x, y)$ combines the vectors $x$ and $y$, whereas in the context of unbinding $\mathcal{B}^*(x, y)$ refers to the retrieval of the vector associated with $y$ from $x$.

$$\mathcal{B}(x, y) = \frac{1}{d} \cdot H(Hx \odot Hy) \qquad (2)$$

Now, we will discuss the binding of $\rho$ different representations, which will become important later in our analysis but is discussed here for adjacency to the binding definition. Composite representation in vector symbolic architectures is defined by the summation of the bound vectors. We define a parameter $\rho \in \mathbb{N} : \rho \geqslant 1$ that denotes the number of vector pairs bundled in a composite representation. Given vectors $x_i, y_i \in \mathbb{R}^d$ and $\forall\, i \in \mathbb{N} : 1 \leqslant i \leqslant \rho$, we can define the composite representation $\chi$ as

$$\underset{\rho=1}{\chi} = \mathcal{B}(x_1, y_1) \qquad \underset{\rho=2}{\chi} = \mathcal{B}(x_1, y_1) + \mathcal{B}(x_2, y_2) \qquad \cdots \qquad \chi_\rho = \sum_{i=1}^{\rho} \mathcal{B}(x_i, y_i) \qquad (3)$$

Next, we require the unbinding operation, which is defined via an inverse function via the following theorem. This will give a symbolic form of our unbinding step that retrieves the original component $\boldsymbol{x}$ being searched for, as well as a necessary noise component $\boldsymbol{\eta}^\circ$, which must exist whenever $\rho \geqslant 2$ items are bound together without expanding the dimension $d$.

**Theorem 3.1** (Inverse Theorem). *Given the identity function $Hx \cdot Hx^\dagger = \mathbb{1}$ where $x^\dagger$ is the inverse of $x$ in Hadamard domain, then $\mathcal{B}^*(\mathcal{B}(x_1, y_1) + \cdots + \mathcal{B}(x_\rho, y_\rho), y_i^\dagger) = \begin{cases} x_i & \text{if} \quad \rho = 1 \\ x_i + \eta_i^\circ & \text{else } \rho > 1 \end{cases}$ where $x_i, y_i \in \mathbb{R}^d$ and $\eta_i^\circ$ is the noise component.*

*Proof of Theorem 3.1.* We start from the identity function $Hx \cdot Hx^\dagger = \mathbb{1}$ and thus $Hx^\dagger = \frac{1}{Hx}$. Now using Equation 2 we get,

$$\mathcal{B}^*(\mathcal{B}(x_1, y_1) + \cdots + \mathcal{B}(x_\rho, y_\rho), y_i^\dagger) = \frac{1}{d} \cdot H((Hx_1 \odot Hy_1 + \cdots + Hx_\rho \odot Hy_\rho) \odot \frac{1}{Hy_i})$$

$$= \frac{1}{d} \cdot H(Hx_i + \frac{1}{Hy_i} \odot \sum_{\substack{j=1 \\ j \neq i}}^{\rho} (Hx_j \odot Hy_j)) = x_i + \frac{1}{d} \cdot H(\frac{1}{Hy_i} \odot \sum_{\substack{j=1 \\ j \neq i}}^{\rho} (Hx_j \odot Hy_j)) \quad Lemma\ 3.1$$

$$= \begin{cases} x_i & \text{if} \quad \rho = 1 \\ x_i + \eta_i^\circ & \text{else } \rho > 1 \end{cases}$$

$\square$

To reduce the noise component and improve the retrieval accuracy, [9, 32] proposes a projection step to the input vectors by normalizing them by the absolute value in the Fourier domain. While such identical normalization is not useful in the Hadamard domain since it will only transform the elements to $+1$ and $-1$s, we will define a projection step with only the Hadamard transformation without normalization given in Definition 3.2.

**Definition 3.2** (Projection). The projection function of $x$ is defined by $\pi(x) = \frac{1}{d} \cdot Hx$.

If we apply the Definition 3.2 to the inputs in Theorem 3.1 then we get

$$\mathcal{B}^*(\mathcal{B}(\pi(x_1), \pi(y_1)) + \cdots + \mathcal{B}(\pi(x_\rho), \pi(y_\rho)), \pi(y_i)^\dagger) = \mathcal{B}^*(\frac{1}{d} \cdot H(x_1 \odot y_1 + \cdots x_\rho \odot y_\rho), \frac{1}{y_i})$$

$$= \frac{1}{d} \cdot H(\frac{1}{y_i} \odot (x_1 \odot y_1 + \cdots x_\rho \odot y_\rho))$$

$$(4)$$

The retrieved value would be projected onto the Hadamard domain as well and to get back the original data we apply the reverse projection. Since the inverse of the Hadamard matrix is the Hadamard matrix itself, in the reverse projection step we just apply the Hadamard transformation again which derives the output to

$$H(\frac{1}{d} \cdot H(\frac{1}{y_i} \odot (x_1 \odot y_1 + \cdots x_\rho \odot y_\rho))) = \frac{1}{y_i} \odot (x_1 \odot y_1 + \cdots + x_\rho \odot y_\rho)$$

$$= \begin{cases} x_i & \text{if} \quad \rho = 1 \\ x_i + \sum_{j=1,\, j \neq i}^{\rho} \frac{x_j y_j}{y_i} & \text{else } \rho > 1 \end{cases} = \begin{cases} x_i & \text{if} \quad \rho = 1 \\ x_i + \eta_i^\pi & \text{else } \rho > 1 \end{cases} \qquad (5)$$

where $\eta_i^\pi$ is the noise component due to the projection step. In expectation, $\eta_i^\pi < \eta_i^\circ$ (see Appendix A). Thus, the projection step diminishes the accumulated noise. More interestingly, the retrieved output

term does not contain any Hadamard matrix. Therefore, we can recast the initial binding definition by multiplying the query vector $y_i$ to the output of Equation 5 which makes the binding function as the sum of the element-wise product of the vector pairs and the compositional structure a linear time $\mathcal{O}(n)$ representation. Thus, the redefinition of the binding function is $\mathcal{B}'(x, y) = x \odot y$ and $\rho$ bundle of the vector pairs is $\chi'_\rho = \sum_{i=1}^{\rho} (x_i \odot y_i)$. Consequently, the unbinding would be a simple element-wise division of the bound representation by the query vector, i.e, $\mathcal{B}^{*'}(x, y) = x \odot \frac{1}{y}$ where $x$ and $y$ are the bound and query vector, respectively.

## 3.1 Initialization of HLB

For the binding and unbinding operations to work, vectors need to have an expected value of zero. However, since we would divide the bound vector with query during unbinding, values close to zero would destabilize the noise component and create numerical instability. Thus, we define a Mixture of $\mathcal{N}$ormal Distribution (MiND) with an expected value of zero but an absolute mean greater than zero given in Equation 6 where $\mathcal{U}$ is the Uniform distribution. Considering half of the elements are sampled for a normal distribution of mean $-\mu$ and the rest of the half with a mean of $\mu$, the resulting vector has a zero mean with an absolute mean of $\mu$. The properties of the vectors sampled from a MiND distribution are given in Properties 3.1.

$$\Omega(\mu, 1/d) = \begin{cases} \mathcal{N}(-\mu, 1/d) & \text{if} \quad \mathcal{U}(0,1) > 0.5 \\ \mathcal{N}(\ \mu, 1/d) & \text{else } \mathcal{U}(0,1) \leqslant 0.5 \end{cases} \tag{6}$$

**Properties 3.1** (Initialization Properties). *Let $x \in \mathbb{R}^d$ sampled from $\Omega(\mu, 1/d)$ holds the following properties.* $\mathrm{E}[x] = 0, \ \mathrm{E}[|x|] = \mu$, *and* $\|x\|_2 = \sqrt{\mu^2 d}$

## 3.2 Similarity Augmentation

In VSAs, it is common to measure the similarity with an extracted embedding $\hat{\boldsymbol{x}}$ with some other vector $\boldsymbol{x}$ using the cosine similarity. For our HLB, we devise a correction term when it is known that $\rho$ items have been bound together to extract $\hat{\boldsymbol{x}}$, i.e., $\mathcal{B}^*(\chi_\rho, \boldsymbol{z}) = \hat{\boldsymbol{x}}$. Then if $\hat{\boldsymbol{x}}$ is the noisy version of the true bound term $\boldsymbol{x}$, we want $\mathrm{cossim}(\hat{\boldsymbol{x}}, \boldsymbol{x}) = 1$, and $\mathrm{cossim}(\hat{\boldsymbol{x}}, \boldsymbol{y}) = 0, \forall \boldsymbol{y} \neq \boldsymbol{x}$. We achieve this by instead computing $\mathrm{cossim}(\hat{\boldsymbol{x}}, \boldsymbol{x}) \cdot \sqrt{\rho}$, and the derivation of this corrective term is given by Theorem 3.2.

**Theorem 3.2** ($\phi - \rho$ Relationship). *Given $x_i, y_i \sim \Omega(\mu, 1/d) \ \forall \ i \in \mathbb{N} : 1 \leqslant i \leqslant \rho$, the cosine similarity $\phi$ between the original $x_i$ and retrieved vector $\hat{x}_i$ is approximately equal to the inverse square root of the number of vector pairs in a composite representation $\rho$ given by $\phi \approx \frac{1}{\sqrt{\rho}}$.*

*Proof of Theorem 3.2.* We start with the definition of cosine similarity and insert the value of $\hat{x}_i$. The step-by-step breakdown is shown in Equation 7.

$$\phi = \frac{\sum\limits^{d} x_i \cdot \hat{x}_i}{\|x_i\|_2 \cdot \|\hat{x}_i\|_2} = \frac{\sum\limits^{d} x_i \cdot \left( x_i + \sum\limits_{j=1, \ j \neq i}^{\rho} \frac{x_j y_j}{y_i} \right)}{\|x_i\|_2 \cdot \|x_i + \sum\limits_{j=1, \ j \neq i}^{\rho} \frac{x_j y_j}{y_i}\|_2} = \frac{\sum\limits^{d} x_i \cdot x_i + \sum\limits^{d} x_i \cdot \left( \sum\limits_{j=1, \ j \neq i}^{\rho} \frac{x_j y_j}{y_i} \right)}{\|x_i\|_2 \cdot \|x_i + \sum\limits_{j=1, \ j \neq i}^{\rho} \frac{x_j y_j}{y_i}\|_2} \tag{7}$$

Employing Properties 3.1 we can derive that $\|x_i\|_2 = \sqrt{\sum x_i \cdot x_i} = \sqrt{\mu^2 d}$ and $\|\frac{x_j y_j}{x_i}\| = \sqrt{\mu^2 d}$.

Thus, the square of the $\|x_i + \sum\limits_{j=1, \ j \neq i}^{\rho} \frac{x_j y_j}{y_i}\|_2$ can be expressed as

$$= \|x_i\|_2^2 + \sum\limits_{j=1, \ j \neq i}^{\rho} \left\| \frac{x_j y_j}{y_i} \right\|_2^2 + 2 \cdot \underbrace{\sum\limits^{d} x_i \left( \sum\limits_{j=1, \ j \neq i}^{\rho} \frac{x_j y_j}{y_i} \right)}_{\alpha} + \underbrace{\sum\limits^{d} \sum\limits_{\substack{j=1 \\ j \neq i}}^{\rho-1} \sum\limits_{\substack{l=1 \\ l \neq j}}^{\rho-1} \frac{x_j y_j}{y_i} \cdot \frac{x_l y_l}{y_i}}_{\beta} \tag{8}$$

$$= \mu^2 d + (\rho - 1) \cdot \mu^2 d + 2\alpha + 2\beta \quad = \rho \cdot \mu^2 d + 2\alpha + 2\beta$$

Therefore, using Equation 7 and Equation 8 we can write that

$$\mathrm{E}[\phi] = \frac{\mu^2 d + \alpha}{\sqrt{\mu^2 d} \cdot \sqrt{\rho \cdot \mu^2 d + 2\alpha + 2\beta}} \approx^1 \frac{\mu^2 d}{\sqrt{\mu^2 d} \cdot \sqrt{\rho \cdot \mu^2 d}} = \frac{\mu^2 d}{\sqrt{\rho} \cdot \mu^2 d} = \frac{1}{\sqrt{\rho}} \qquad \square$$

The experimental result of the $\phi - \rho$ relationship closely follows the theoretical expectation provided in Appendix C which also indicates that the approximation is valid. Since, we know from Theorem 3.2 that similarity score $\phi$ drops by the inverse square root of the number of vector pairs in a composite representation $\rho$, in places where $\rho$ is known or can be estimated from $\|\chi_\rho\|_2 \approx \mu^2 \sqrt{\rho \cdot d}$ (proof in Appendix B), it can be used to update the cosine similarity multiplying the scores by $\sqrt{\rho}$. Equation 9 shows the updated similarity score where in a positive case $(+)$, $\phi$ would be close to $1/\sqrt{\rho}$ and in a negative case $(-)$, $\phi$ would be close to zero.

$$\phi' = \phi \times \sqrt{\rho} \qquad \phi'_{(+)} = \phi_{\to \frac{1}{\sqrt{\rho}}} \times \sqrt{\rho} \approx 1 \qquad \phi'_{(-)} = \phi_{\to 0} \times \sqrt{\rho} \approx 0 \tag{9}$$

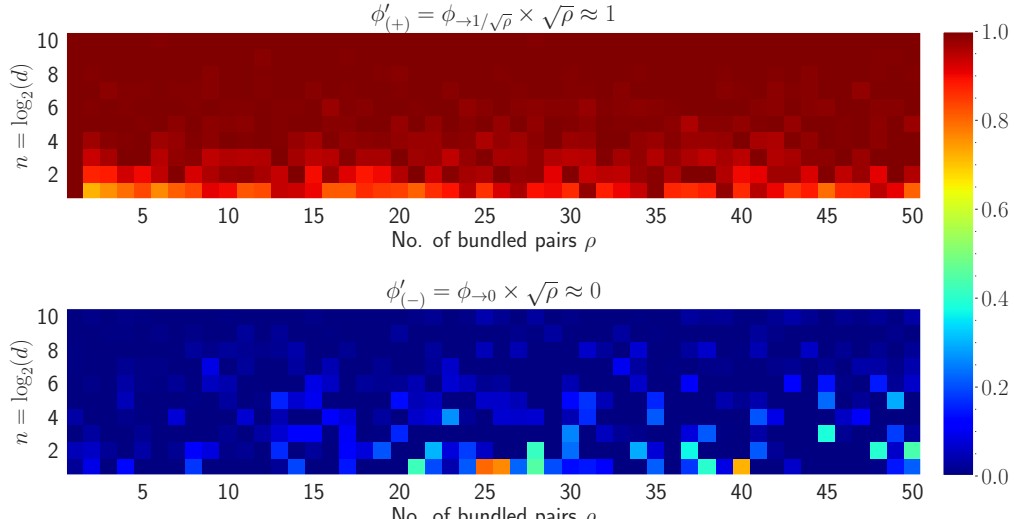

Figure 1: Empirical comparison of the corrected cosine similarity scores between $\phi'_{(+)}$ (on top) and $\phi'_{(-)}$ (on bottom) for varying $n$ and $\rho$ shown in heatmap. The dimension, i.e., $d = 2^n$ is varied from 2 to 1024 ($n \in \{1, 2, \cdots, 10\}$) and the number of vector pairs bundled is varied from 1 to 50. This shows that we can accurately identify when a vector $x$ has been bound to a VSA or not when we keep track of how many pairs of terms $\rho$ are included.

Empirical results of $\phi'$ for varying $n$ and $\rho$ are visualized and verified by a heatmap. In a composite representation $\chi'_\rho = \sum_{i=1}^{\rho} (x_i \odot y_i)$, when unbinding is applied using the query $y_i$, i.e., $\mathcal{B}^{*'}(\chi'_\rho, y_i) = \hat{x}_i$, a positive case is a similarity between $x_i$ and $\hat{x}_i$. On the contrary, similarity between $\hat{x}_i$ and any $x_j$ where $j \in \{1, 2, \cdots, \rho\}$ and $j \neq i$, is a negative case. Mean cosine similarity scores of 100 trials for both positive and negative cases in presented in Figure 1 where the scores for the positive cases are in the red ($\approx 1$) shades and the scores for the negative cases are in the blue ($\approx 0$) shades.

## 4 Empirical Results

### 4.1 Classical VSA Tasks

A common VSA task is, given a bundle (addition) of $\rho$ pairs of bound vectors $s = \sum_{i=1}^{\rho} \mathcal{B}(x_i, y_i)$, given a query $x_q \in s$, c can the corresponding vector $y_q$ be correctly retrieved from the bundle. To test this, we perform an experiment similar to one in [37]. We first create a pool $P$ of $N = 1000$ random vectors, then sample (with replacement) $p$ pairs of vectors for $p \in \{1, 2, \cdots, 25\}$. The pairs are bound together and added to create a composite representation $s$. Then, we iterate through all left pairs $x_q$ in the composite representation and attempt to retrieve the corresponding $y_q, \forall q \in [1, p]$. A retrieval is considered correct if $\mathcal{B}^*(s, x_q)^\top y_q > \mathcal{B}^*(s, x_q)^\top y_j, \forall j \neq q$. The total accuracy score for the bundle is recorded, and the experiment is repeated for 50 trials. Experiments are performed to

---

[1]Here, $\alpha$ and $\beta$ are the noise terms and in expectation $\mathrm{E}[\alpha] \approx 0$ and $\mathrm{E}[\beta] \approx 0$.

compare HRR [32], VTB [12], MAP [10], and our HLB VSAs. For each VSA, at each dimension of the vector, the area under the curve (AUC) of the accuracy vs. the no. of bound terms plot is computed, and the results are shown in Figure 2. In general, HLB has comparable performance to HRR and VTB, and performs better than MAP.

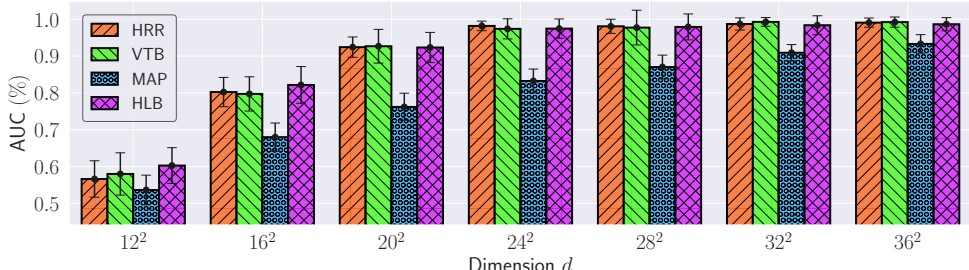

Figure 2: The area under the accuracy curve due to the change of no. of bundled pairs $\rho$ for dimensions $d$. All the dimensions are chosen to be perfect squares due to the constraint of VTB.

The scenario we just considered looked at bindings of only two items together, summed of many pairs of bindings. [12] proposed addition evaluations over sequential bindings that we now consider. In the *random* case we have an initial vector $b_0$, and for $p$ rounds, we will modify it by a random vector $x_t$ such that $b_{t+1} = \mathcal{B}(b_t, x_t)$, after which we unbind each $x_t$ to see how well the previous $b_t$ is recovered. In the *auto binding* case, we use a single random vector $x$ for all $p$ rounds.

In this task, we are concerned with the quality of the similarity score in random/auto-binding, as we want $\mathcal{B}^*(b_{t+1}, x_t)^\top b_t = 1$. For VSAs with approximate unbinding procedures, such as HRR, VTB, and MAP-C, the returned value will be 1 if $p = 1$ but will decay as $p$ increases. HLBuses an exact unbinding procedure so that the returned value is expected to be $1 \forall p$. We are also interested in the magnitude of the vectors $\|\mathcal{B}^*(b_{t+1}, x_t)\|_2$, where an ideal VSA has a constant magnitude that does not explode/vanish as $p$ increases.

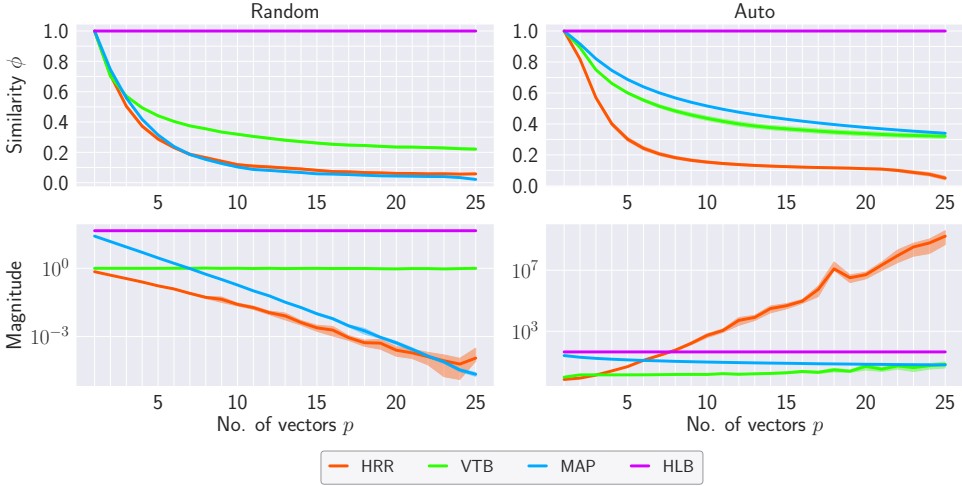

Figure 3: When repeatedly binding different random (left) or a single vector (right), HLB consistently returns the ideal similarity score of 1 for a present item (top row) and has a constant magnitude (bottom row), avoiding exploding/vanishing values.

Figure 3 shows that HLB maintains a stable magnitude regardless of the number of bound vectors in both cases. This property arises due to the properties of the distribution shown in Properties 3.1. As all components have an expected absolute value of 1, the product of all components also has an expected absolute value of 1. Thus, the norm of the binding is simply $\sqrt{d}$. It also shows HLB maintains the desired similarity score as $p$ increases. Combined with Figure 1 that shows the scores are near-zero when an item is not present, HLB has significant advantages in consistency for designing VSA solutions.

## 4.2 Deep Learning with Hadamard-derived Linear Binding

Two recent methods that integrate HRR with deep learning are tested to further validate our approach and briefly summarized in the two sub-sections below. In each case, we run all four VSAs and see that HLB either matches or exceeds the performance of other VSAs. In every experiment, the standard method of sampling vectors from each VSA is followed as outlined in Table 1. All the experiments are performed on a single NVIDIA TESLA PH402 GPU with 32GB memory.

### 4.2.1 Connectionist Symbolic Pseudo Secrets

When running on low-power computing environments, it is often desirable to offload the computation to a third-party cloud environment to get the answer faster and use fewer local resources. However, this may be problematic if one does not fully trust the available cloud environments. Homomorphic encryption (HE) is the ideal means to alleviate this problem, providing cryptography for computations. HE is currently more expensive to perform than running a neural network itself [11], defeating its own utility in this scenario. Connectionist Symbolic Pseudo Secrets (CSPS) [3] provides a heuristic means of obscuring the nature of the input (content), and output (number of classes/prediction), while also reducing the total local compute required.

CSPS mimics a "one-time-pad" by taking a random VSA vector $s$ as the *secret* and binding it to the input $x$. The value $\mathcal{B}(s, x)$ obscures the original $x$, and the third-party runs the bulk of the network on their platform. A result $\tilde{y}$ is returned, and a small local network computes the final answer after unbinding with the secret $\mathcal{B}^*(\tilde{y}, s)$. Other than changing the VSA used, we follow the same training, testing, architecture size, and validation procedure of [3].

CSPS experimented with 5 datasets MNIST, SVHN, CIFAR-10 (CR10), CIFAR-100 (CR100), and Mini-ImageNet (MIN). First, we look at the accuracy of each method, which is lower due to the noise of the random vector $s$ added at test time since no secret VSA is ever reused. The results are shown in Table 2, where HLB outperforms all prior methods significantly. Notably, the MAP VSA is second best despite being one of the older VSAs, indicating its similarity to HLB in using a simple binding procedure, and thus simple gradient may be an important factor in this scenario.

Table 2: Accuracy comparison of the proposed HLB with HRR, VTB, MAP-C, and MAP-B in CSPS. The dimensions of the inputs along with the no. of classes are listed in the Dims/Labels column. The last row shows the geometric mean of the results.

| Dataset | Dims/ Labels | CSPS + HRR | | CSPS + VTB | | CSPS + MAP-C | | CSPS + MAP-B | | CSPS + HLB | |
|---|---|---|---|---|---|---|---|---|---|---|---|
| | | Top@1 | Top@5 | Top@1 | Top@5 | Top@1 | Top@5 | Top@1 | Top@5 | Top@1 | Top@5 |
| MNIST | $28^2/10$ | 98.51 | – | 98.44 | – | 98.46 | – | 98.40 | – | **98.73** | – |
| SVHN | $32^2/10$ | 88.44 | – | 19.59 | – | 79.95 | – | 92.43 | – | **94.53** | – |
| CR10 | $32^2/10$ | 78.21 | – | 74.22 | – | 76.69 | – | 82.83 | – | **83.81** | – |
| CR100 | $32^2/100$ | 48.84 | 75.82 | 35.87 | 61.79 | 56.77 | 81.52 | 57.76 | 84.63 | **58.82** | **87.50** |
| MIN | $84^2/100$ | 40.99 | 66.99 | 45.81 | 73.52 | 52.22 | 78.63 | 57.91 | 82.81 | **59.48** | **83.35** |
| GM | | 67.14 | 71.26 | 47.24 | 67.40 | 70.89 | 80.06 | 75.90 | 83.72 | **77.17** | **85.40** |

However, improved accuracy is not useful in this scenario if more information is leaked. The test in this scenario, as proposed by [3], is to calculate the Adjusted Rand Index (ARI) after attempting to cluster the inputs $x$ and the outputs $\hat{y}$, which are available/visible to the snooping third-party. To be successful, the ARI must be near zero (indicating random label assignment) for both inputs and outputs.

We use K-means, Gaussian Mixture Model (GMM), Birch [45], and HDBSCAN [7] as the clustering algorithms and specify the true number of classes to each method to maximize attacker success (information they would not know). The results can be found in Table 3, where the top rows indicate the clustering of the input $\mathcal{B}(x, s)$, and the bottom rows the clustering of the output $\hat{y}$. All the numbers are percentages $(\%)$, showing all methods do a good job at hiding information from the adversary (except on the MNIST dataset, which is routinely degenerate).

The MNIST result is a good reminder that CSPS security is heuristic, not guaranteed. Nevertheless, we see HLB has consistently close-to-zero scores for SVHN, CIFARs, and Mini-ImageNet, indicating

Table 3: Clustering results of the main network inputs (top rows) and outputs (bottom rows) in terms of Adjusted Rand Index (ARI). Because CSPS is trying to hide information, scores near zero are better. Cell color corresponds to the cell absolute value, with blue indicating lower ARI and red indicating higher ARI. All numbers in percentages, and show HLB is better at information hiding.

| CLUSTERING METHODS | HRR | | | | | VTB | | | | |
|---|---|---|---|---|---|---|---|---|---|---|
| | MNIST | SVHN | CR10 | CR100 | MIN | MNIST | SVHN | CR10 | CR100 | MIN |
| K-MEANS | −0.02 | −0.01 | 0.18 | 0.54 | 0.42 | −0.00 | −0.01 | −0.01 | 0.02 | 0.00 |
| GMM | 0.01 | 0.00 | 0.09 | 0.61 | 0.44 | 4.67 | 1.37 | −0.01 | 0.02 | 0.01 |
| BIRCH | 0.20 | 0.00 | 0.14 | 0.45 | 0.35 | 0.02 | 0.03 | 0.04 | 0.08 | 0.03 |
| HDBSCAN | 0.00 | −0.24 | 1.23 | 0.01 | 0.02 | 0.00 | 0.00 | 0.00 | 0.00 | 0.00 |
| K-MEANS | 1.28 | 0.06 | 0.21 | 0.03 | 0.08 | 8.52 | 0.13 | 1.11 | 0.05 | 0.12 |
| GMM | 1.28 | 0.06 | 0.17 | 0.04 | 0.09 | 8.63 | 0.14 | 1.63 | 0.05 | 0.00 |
| BIRCH | 1.51 | 0.03 | 0.13 | 0.05 | 0.07 | 3.24 | 0.00 | 0.64 | 0.06 | 0.17 |
| HDBSCAN | 0.00 | 0.00 | 0.00 | 0.00 | 0.00 | 0.00 | 0.09 | 0.00 | 0.00 | 0.00 |

| CLUSTERING METHODS | MAP | | | | | HLB | | | | |
|---|---|---|---|---|---|---|---|---|---|---|
| | MNIST | SVHN | CR10 | CR100 | MIN | MNIST | SVHN | CR10 | CR100 | MIN |
| K-MEANS | 0.17 | 0.01 | 0.01 | 0.00 | 0.00 | 0.09 | 0.00 | 0.00 | 0.00 | 0.00 |
| GMM | 3.39 | −0.01 | 0.01 | 0.00 | 0.00 | 2.53 | 0.00 | 0.00 | 0.00 | 0.00 |
| BIRCH | 0.84 | −0.00 | 0.00 | 0.01 | 0.00 | 0.83 | 0.00 | 0.00 | 0.01 | 0.00 |
| HDBSCAN | 0.00 | 0.00 | 0.00 | 0.00 | 0.00 | 0.00 | 0.00 | 0.00 | 0.00 | 0.00 |
| K-MEANS | 15.91 | 0.09 | 0.00 | 0.03 | 0.01 | 13.67 | −0.04 | 0.01 | 0.02 | −0.00 |
| GMM | 42.43 | 0.11 | 0.00 | 0.03 | 0.00 | 14.96 | −0.04 | 0.01 | 0.02 | 0.00 |
| BIRCH | 7.09 | −0.07 | −0.02 | 0.01 | −0.00 | 18.44 | −0.07 | 0.00 | 0.01 | 0.02 |
| HDBSCAN | 0.48 | 0.00 | 0.00 | 0.00 | 0.00 | 7.60 | 0.01 | 0.00 | 0.00 | 0.00 |

that its improved accuracy with simultaneously improved security. This also validates the use of the VSA in deep learning architecture design and the efficacy of our approach.

### 4.2.2 Xtreme Multi-Label Classification

Extreme Multi-label (XML) is the scenario where, given a single input of size $d$, $C >> d$ classes are used to predict. This is common in e-commerce applications where new products need to be tagged, and an input on the order of $d \approx 5000$ is relatively small compared to $C \geqslant 100,000$ or more classes. This imposes unique computational constraints due to the output space being larger than the input space and is generally only solvable because the output space is sparse — often less than 100 relevant classes will be positive for any one input. VSAs have been applied to XML by exploiting the low positive class occurrence rate to represent the problem symbolically [9].

While many prior works focus on innovative strategies to cluster/make hierarchies/compress the penultimate layer[19, 20, 31, 18, 44, 22], a neuro-symbolic approach was proposed by [9]. Given $K$ total possible classes, they assigned each class a vector $c_k$ to be each class's representation, and the set of all classes $a = \sum_{k=1}^{K} c_k$.

The VSA trick used by [9] was to define an additional "present" class $p$ and a "missing" class $m$. Then, the target output of the network $f(\cdot)$ is itself a vector composed of two parts added together. First, $\mathcal{B}(p, \sum_k c_k)$ represents all *present* classes, and so the sum is over a finite smaller set. Then the absent classes compute the *missing* representing $\mathcal{B}(m, a - \sum_k c_k)$, which again only needs to compute over the finite set of present classes, yet represents the set of all non-present classes by exploiting the symbolic properties of the VSA.

For XML classification, we have a set of $K$ classes that will be present for a given input, where $K \approx 10$ is the norm. Yet, there will be $L$ total possible classes where $L \geqslant 100,000$ is quite common. Forming a normal linear layer to produce $L$ outputs is the majority of computational work and memory use in standard XML models, and thus the target for reduction. A VSA can be used to side-step this cost, as shown by [9], by leveraging the symbolic manipulation of the outputs. First, consider the target label as a vector $s \in \mathbb{R}^d$ such that $d \ll L$. By defining a VSA vector to represent "present" and "missing" classes as $\mathbf{p}$ and $\mathbf{m}$, where each class is given its own vector $c_{1,...,L}$, we can shift the computational complexity form $\mathcal{O}(L)$ to $\mathcal{O}(K)$ by manipulating the "missing" classes as the compliment of the present classes as shown in Equation 10.

Similarly, the loss to calculate the gradient can be computed based on the network's prediction $\hat{s}$ by taking the cosine similarity between each expected class and one cosine similarity for the representation of all missing classes. The excepted response of 1 or 0 for an item being present/absent from the VSA is used to determine if we want the similarity to be 0 (1-cos) or 1 (just cos), as shown in Equation 11.

$$s = \overbrace{\sum_{i \in y_i = 1} \mathcal{B}(\boldsymbol{p}, \boldsymbol{c}_i)}^{\text{Labels Present} \mathcal{O}(dK)} + \overbrace{\sum_{j \in y_j = -1} \mathcal{B}(\boldsymbol{m}, \boldsymbol{c}_j)}^{\text{Labels Absent} \mathcal{O}(dL)} = \overbrace{\mathcal{B}\left(\boldsymbol{p}, \left(\boldsymbol{a} =: \sum_{i \in y_i = 1} \boldsymbol{c}_i\right)\right)}^{\text{Labels Present} \mathcal{O}(d\ K)} + \overbrace{\mathcal{B}\left(\boldsymbol{m}, \left(\boldsymbol{a} - \sum_{i \in y_i = 1} \boldsymbol{c}_i\right)\right)}^{\text{Labels Absent} \mathcal{O}(dK)} \tag{10}$$

$$loss = \overbrace{\sum_{i \in y_i = 1} \left(1 - \cos\left(\mathcal{B}^*(\boldsymbol{p}, \hat{s}), \boldsymbol{c}_i\right)\right)}^{\text{Present Classes } \mathcal{O}(d\ K)} + \overbrace{\cos\left(\mathcal{B}^*(\boldsymbol{m}, \hat{s}), \sum_{i \in y_i = 1} \boldsymbol{c}_i\right)}^{\text{Absent classes } O(d\ K)} \tag{11}$$

The details and network sizes of [9] are followed, except we replace the original VSA with our four candidates. The network is trained on 8 datasets listed in Table 4 from [4] and evaluated using normalized discounted cumulative gain (nDCG) and propensity-scored (PS) based normalized discounted cumulative gain (PSnDCG) as suggested by [19].

Table 4: XML classification results in dense label representation with HRR, VTB, MAP, and HLB in terms of nDCG and PSnDCG. The proposed HLB has attained the best nDCG and PSnDCG scores on all the datasets setting a new SOTA.

| DATASET | BIBTEX | | DELICIOUS | | MEDIAMILL | | EURLEX-4K | |
|---|---|---|---|---|---|---|---|---|
| METRICS | nDCG | PSnDCG | nDCG | PSnDCG | nDCG | PSnDCG | nDCG | PSnDCG |
| HRR | 60.296 | 45.572 | 66.454 | 30.016 | 83.885 | 63.684 | 77.225 | 30.684 |
| VTB | 57.693 | 45.219 | 63.325 | 31.449 | 87.232 | 66.948 | 76.964 | 31.180 |
| MAP-C | 59.280 | 46.092 | 65.376 | 31.943 | 87.255 | 66.886 | 72.439 | 26.752 |
| MAP-B | 59.412 | 46.340 | 65.431 | 32.122 | 86.886 | 66.562 | 71.128 | 26.340 |
| HLB | **61.741** | **48.639** | **67.821** | **32.797** | **88.064** | **67.525** | **77.868** | **31.526** |
| DATASET | EURLEX-4.3K | | WIKI10-31K | | AMAZON-13K | | DELICIOUS-200K | |
| METRICS | nDCG | PSnDCG | nDCG | PSnDCG | nDCG | PSnDCG | nDCG | PSnDCG |
| HRR | 84.497 | 38.545 | 81.068 | 9.185 | 93.258 | 49.642 | 44.933 | 6.839 |
| VTB | 84.663 | 38.540 | 78.025 | 9.645 | 92.373 | 49.463 | 44.092 | 6.664 |
| MAP-C | 85.472 | 39.233 | 80.203 | 10.027 | 92.013 | 48.686 | 45.373 | 6.862 |
| MAP-B | 85.023 | 38.820 | 80.238 | 10.035 | 92.307 | 48.812 | 45.459 | 6.870 |
| HLB | **88.204** | **43.622** | **83.589** | **11.869** | **93.672** | **50.270** | **46.331** | **6.952** |

The classification result in terms of nDCG and PSnDCG in all the eight datasets is presented in Table 4 where the top four datasets are comparatively easy with maximum no. of features of 5000 and no. of labels of 4000. The bottom four datasets are comparatively hard with the no. of features and labels on the scale of $100K$. The proposed HLB has attained the best results in all the datasets on both metrics. In contrast to the prior CSPS results, here we see that the performance differences between HRR, VTB, and MAP are more varied, with no clear "second-place" performer.

## 5   Conclusion

In this paper, a novel linear vector symbolic architecture named HLB is presented derived from Hadamard transform. Along with an initialization condition named MiND distribution is proposed for which we proved the cosine similarity $\phi$ is approximately equal to the inverse square root of the no. of bundled vector pairs $\rho$ which matches with the experimental results. The proposed HLB showed superior performance in classical VSA tasks and deep learning compared to other VSAs such as HRR, VTB, and MAP. In learning tasks, HLB is applied to CSPS and XML classification tasks. In both of the tasks, HLB has achieved the best results in terms of respective metrics in all the datasets showing a diverse potential of HLB in Neuro-symbolic AI.

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

# A  Noise Decomposition

When a single vector pair is combined, one of the vector pairs can be exactly retrieved with the help of the other component and the inverse function, recalling the retrieved output does not contain any noise component for a single pair of vectors, i.e., $\rho = 1$. However, when more than one vector pairs are bundled, noise starts to accumulate. In this section, we will uncover the noise components accumulated with and without the projection to the inputs and analyze their impact on expectation. We first start with the noise component without the projection step $\eta_i^\circ$.

$$\eta_i^\circ = \frac{1}{d} \cdot H\left(\frac{1}{Hy_i} \odot \sum_{\substack{j=1 \\ j \neq i}}^{\rho} (Hx_j \odot Hy_j)\right) \tag{12}$$

Let, set the value of $n$ to be 1 thus, $d = 2^n = 2$ and the number of vector pairs $\rho = 2$, i.e., $\chi_{\rho=2} = \mathcal{B}(x_1, y_1) + \mathcal{B}(x_2, y_2)$. We want to retrieve $x_1$ using the query $y_1$, thereby, the expression of $\eta_i^\circ$ is uncovered step by step for $\rho = 2$ shown in Equation 13.

$$
\begin{aligned}
\eta_i^\circ \Big|_{\rho=2} &= \frac{1}{d} \cdot H\left(\frac{1}{Hy_1} \odot (Hx_2 \odot Hy_2)\right) \\
&= \frac{1}{d} \cdot \sqrt{d} \cdot H\left(\begin{array}{c} \frac{1}{y_1^{(0)}+y_1^{(1)}} \\ \frac{1}{y_1^{(0)}-y_1^{(1)}} \end{array} \odot \begin{array}{c} (x_2^{(0)} + x_2^{(1)}) \cdot (y_2^{(0)} + y_2^{(1)}) \\ (x_2^{(0)} - x_2^{(1)}) \cdot (y_2^{(0)} - y_2^{(1)}) \end{array}\right) \\
&= \frac{1}{d} \cdot \left(\begin{array}{c} \frac{(x_2^{(0)}+x_2^{(1)})(y_2^{(0)}+y_2^{(1)})(y_1^{(0)}-y_1^{(1)}) + (x_2^{(0)}-x_2^{(1)})(y_2^{(0)}-y_2^{(1)})(y_1^{(0)}+y_1^{(1)})}{(y_1^{(0)}+y_1^{(1)})(y_1^{(0)}-y_1^{(1)})} \\ \frac{(x_2^{(0)}+x_2^{(1)})(y_2^{(0)}+y_2^{(1)})(y_1^{(0)}-y_1^{(1)}) - (x_2^{(0)}-x_2^{(1)})(y_2^{(0)}-y_2^{(1)})(y_1^{(0)}+y_1^{(1)})}{(y_1^{(0)}+y_1^{(1)})(y_1^{(0)}-y_1^{(1)})} \end{array}\right) \\
&= \left(\begin{array}{c} \frac{\varphi_1}{\prod_{k=1}^{d}(Hy_1)_k} \\ \frac{\varphi_2}{\prod_{k=1}^{d}(Hy_1)_k} \end{array}\right) \\
&= \frac{\mathcal{P}(x_2, y_2, y_1)}{\prod_{k=1}^{d}(Hy_1)_k}
\end{aligned} \tag{13}
$$

Here, $\varphi_k \ \forall \ k \in \mathbb{N} : 1 \leqslant k \leqslant d$ are the polynomials comprises of $(x_2, \ y_2)$, and the query vector $y_1$. $\mathcal{P}$ is the vector of polynomials consisting of $\varphi_k$. From the noise expression, we can observe that the numerator is a polynomial and the denominator is the product of all the elements of the Hadamard transformation of the query vector. This is true for any value of $n$ and $\rho$. Thus, in general, for any query $y_i$ we can express $\eta_i^\circ$ as shown in Equation 14.

$$\eta_i^\circ = \frac{\overset{\rho}{\underset{j=1, \ j\neq i}{\mathcal{P}}}(x_j, y_j, y_i)}{\prod_{k=1}^{d}(Hy_i)_k} \tag{14}$$

The noise accumulated after applying the projection to the inputs is quite straightforward as given in Equation 15.

$$\eta_i^\pi = \frac{\sum_{j=1, \ j\neq i}^{\rho}(x_j \odot y_j)}{y_i} \tag{15}$$

Although the vectors $x_i, y_i \ \forall \ i \in \mathbb{N} : 1 \leqslant i \leqslant \rho$ are sampled from a MiND with an expected value of 0 given in Equation 6, the sample mean of $x_i$ or $y_i$ would be $\hat{\mu} \approx 0$ but $\hat{\mu} \neq 0$. Both the numerator of $\eta_i^\circ$ and $\eta_i^\pi$ are the polynomials thus the expected value would be very close to 0. However, the expected value of the denominator of $\eta_i^\circ$ would be $\mathrm{E}[\prod_{k=1}^{d}(Hy_i)_k] = \prod_{k=1}^{d}\mathrm{E}[(Hy_i)_k] = \hat{\mu}^d$ whereas the expected value of the denominator of $\eta_i^\pi$ is $\mathrm{E}[y_i] = \hat{\mu}$. Since, $\hat{\mu}^d < \hat{\mu}$, hence, in

expectation $\eta_i^\pi < \eta_i^\circ$. This is also verified by an empirical study where $n$, i.e., the dimension $d = 2^n$ is varied along with the no. of bound vector pairs $\rho$ and the amount of absolute mean noise in retrieval is estimated.

Figure 4 shows the heatmap visualization of the noise for both $\eta_i^\circ$ and $\eta_i^\pi$ in natural log scale. The amount of noise accumulated without any projection to the inputs is much higher compared to the noise accumulation with the projection. For varying $n$ and $\rho$, the maximum amount of noise accumulated when projection is applied is 7.18 and without any projection, the maximum amount of noise is 19.38. Also, most of the heatmap of $\eta_i^\pi$ remains in the blue region whereas as $n$ and $\rho$ increase, the heatmap of $\eta_i^\circ$ moves towards the red region. Therefore, it is evident that the projection to the inputs diminishes the amount of accumulated noise with the retrieved output.

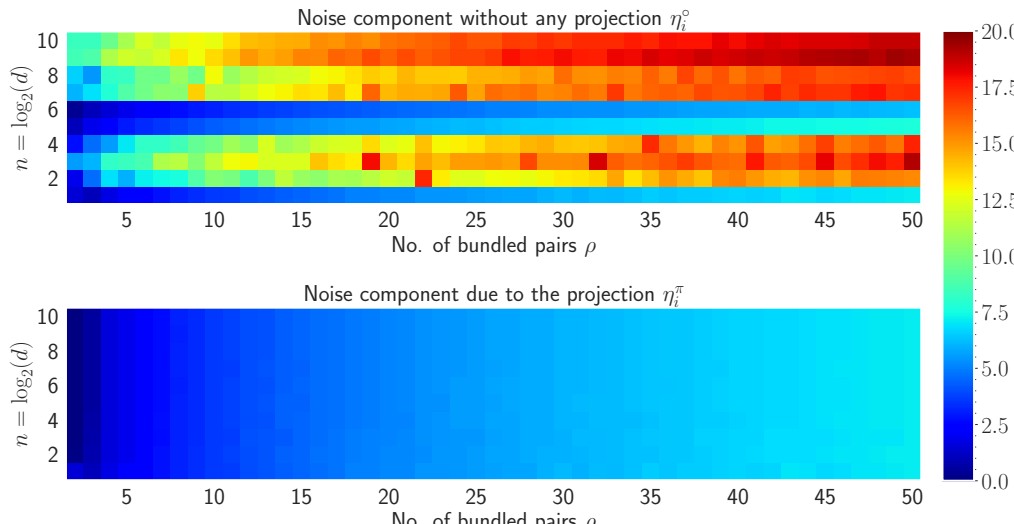

Figure 4: Heatmap of the empirical comparison of the noise components $\eta_i^\circ$ and $\eta_i^\pi$ for varying $n$ and $\rho$ shown in natural logarithm scale. The dimension, i.e., $d = 2^n$ is varied from 2 to 1024 ($n \in \{1, 2, \cdots, 10\}$) and the number of vector pairs bundled is varied from 2 to 50.

# B  Norm Relation

**Theorem B.1** ($\chi_\rho - \rho$ Relationship). *Given $x_i, y_i \sim \Omega(\mu, 1/d) \in \mathbb{R}^d \; \forall \; i \in \mathbb{N} : 1 \leqslant i \leqslant \rho$, the norm of the composite representation $\chi_\rho$ is proportional to $\sqrt{\rho}$ and approximately equal to the $\mu^2 \sqrt{\rho \cdot d}$.*

*Proof of Theorem B.1.* Given $\chi_\rho$ is the composite representation of the bound vectors, i.e., the summation of $\rho$ no. of individual bound terms. First, let's compute the norm of the single bound term as shown in Equation 16.

$$
\begin{aligned}
\|\mathcal{B}(x_i, y_i)\|_2 &= \|x_i \cdot y_i\|_2 \\
&= \sqrt{(x_i^{(1)} y_i^{(1)})^2 + (x_i^{(2)} y_i^{(2)})^2 + \cdots + (x_i^{(d)} y_i^{(d)})^2} \\
&= \sqrt{(\pm\mu^2)^2 + (\pm\mu^2)^2 + \cdots + (\pm\mu^2)^2} \quad \left[E[x^{(1)}] \cdot E[y^{(1)}] = \pm\mu \cdot \pm\mu = \pm\mu^2\right] \\
&= \sqrt{\mu^4 d}
\end{aligned}
$$

$$(16)$$

Now, let's expand and compute the square norm of the composite representation given in Equation 17.

$$\left\|\chi_\rho\right\|_2^2 = \left\|\mathcal{B}(x_1, y_1) + \mathcal{B}(x_2, y_2) + \cdots + \mathcal{B}(x_\rho, y_\rho)\right\|_2^2$$

$$= \left\|\mathcal{B}(x_1, y_1)\right\|_2^2 + \left\|\mathcal{B}(x_2, y_2)\right\|_2^2 + \cdots + \left\|\mathcal{B}(x_\rho, y_\rho)\right\|_2^2 + \xi$$

where $\xi$ is the rest of the terms of square expansion.

$$= \mu^4 d + \mu^4 d + \cdots + \mu^4 d + \xi \tag{17}$$

$$= \rho \cdot \mu^4 d + \xi$$

$$\left\|\chi_\rho\right\|_2 = \sqrt{\rho \cdot \mu^4 d + \xi}$$

$$\approx \sqrt{\rho \cdot \mu^4 d} \quad [\ \xi \text{ is the noise term and discarded to make an approximation } ]$$

$$= \mu^2 \sqrt{\rho \cdot d} \quad \square$$

Thus, given the composite representation and the mean of the MiND distribution, we can estimate the no. of bound terms bundled together by $\rho \approx \left\|\chi_\rho\right\|_2^2 / \mu^4 d$. $\qquad\square$

Figure 5 shows the comparison between the theoretical relationship and actual experimental results where the norm of the composite representation is computed for $\mu = 0.5$ and $\rho = \{1, 2, \cdots, 200\}$. The figure indicates that the theoretical relationship aligns with the experimental results. However, as the number of bundled pair increases, the variation in the norm increases. This is because of making the approximation by discarding $\xi$ in Equation 17.

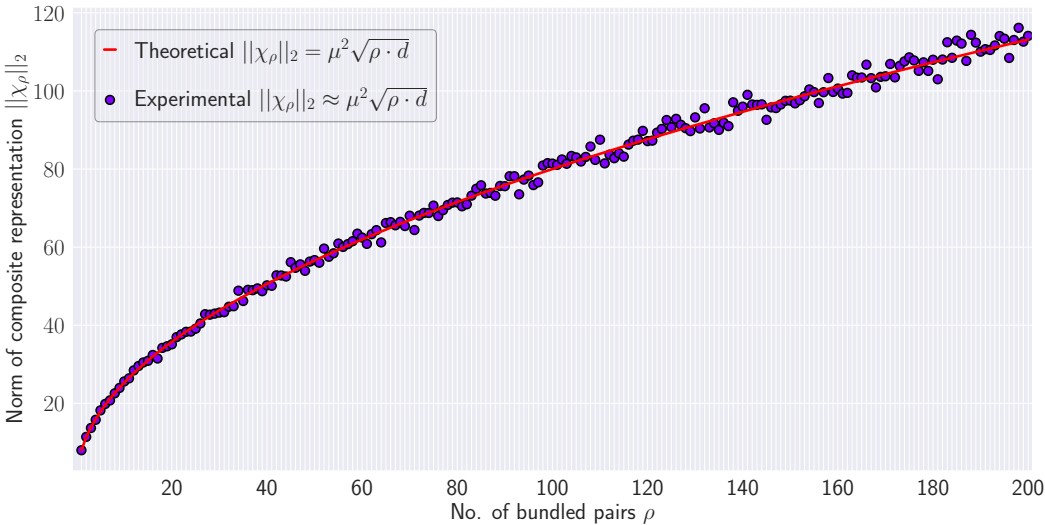

Figure 5: Comparison between the theoretical and experimental relationship of Theorem B.1. The norm of the composite representation of the bound vectors is computed for no. of bundled vectors from 1 to 200 of dimension $d = 1024$. The figure shows how the experimental value of the norm closely follows the theoretical relation between $\left\|\chi_\rho\right\|_2$ and $\rho$.

# C   Cosine Relation

Theorem 3.2 shows how the cosine similarity $\phi$ between the original $x_i$ and retrieved vector $\hat{x}_i$ is approximately equal to the inverse square root of the number of vector pairs in a composite representation $\rho$. In this section, we will perform an empirical analysis of the theorem and compare it with the theoretical results. For $\rho = \{1, 2, \cdots, 50\}$, similarity score $\phi$ is calculated for vector dimension $d = 512$. Additionally, the theoretical cosine similarity score is also calculated using the value of $\phi$ following the theorem. Figure 6 shows the comparison between the two results where the experimental result closely follows the theoretical result. The figure also shows the standard deviation for 100 trials, indicating a minute change from the actual value.

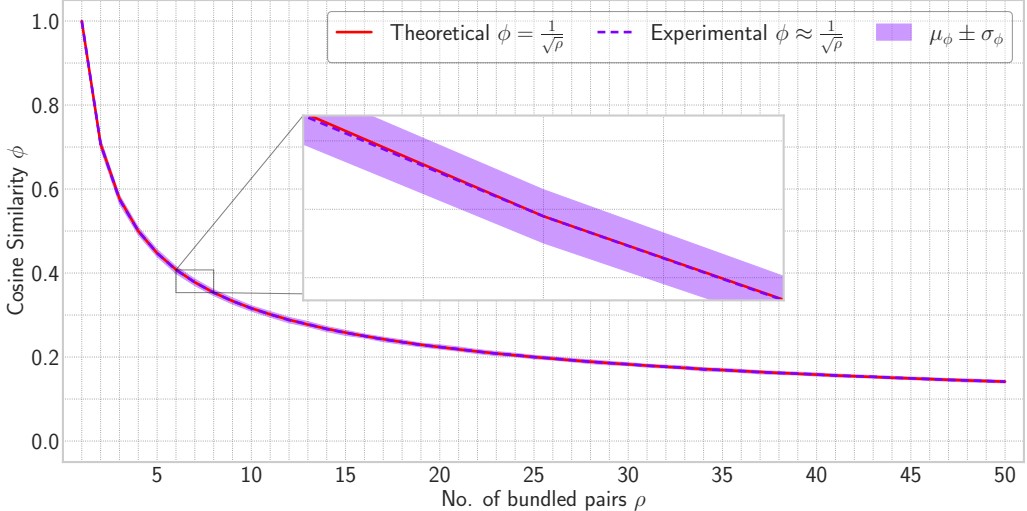

Figure 6: Comparison between the theoretical and experimental $\phi - \rho$ relationship. Vectors of dimension $d = 512$ are combined and retrieved with a varied number of vectors from 1 to 50. The zoom portion shows how closely experimental results match with the theoretical conclusion.

