# OpenReview forum: "A Walsh Hadamard Derived Linear Vector Symbolic Architecture"
_NeurIPS.cc/2024/Conference — NeurIPS 2024 poster_

### Official Review · Reviewer_FSGF · 2024-06-23

**Soundness:** 3
**Presentation:** 3
**Contribution:** 3
**Rating:** 7
**Confidence:** 3

**Summary:**

The paper introduces a new Vector Symbolic Architecture (VSA) termed Hadamard-derived Linear Binding (HLB), aimed at enhancing computational efficiency and performance in both classical VSA tasks and deep learning applications. VSAs involve binding two vectors to create a new vector in the same space, supporting symbolic operations like binding and unbinding, which is crucial for neuro-symbolic AI.

Traditional VSAs, such as Tensor Product Representation (TPR) and Holographic Reduced Representation (HRR), have limitations like computational complexity or numerical stability issues. In contrast, HLB leverages the Hadamard transform, ensuring linear complexity for binding and addressing these drawbacks.

**Strengths:**

- The paper is well organized and the concepts are clearly explained.
- The paper details theoretical foundations, derivation processes, and empirical evaluations demonstrating HLB's efficacy across various tasks. It concludes with implications for both classical symbolic AI and modern deep learning systems.

**Weaknesses:**

I do not observe a clear weakness

**Questions:**

Since I am not familiar with this VSA domain, I'm eager to understand the problem setting in the experiment sections 4.2.1 and 4.2.2. Could the author elaborate with simple example on these tasks?

**Limitations:**

The work has no negative societal impacts.

---

> ### Author Rebuttal · Authors · 2024-08-06
>
> We will add the explanations below to the manuscript to provide motivating reasons for each approach.
>
> **4.2.1:**
> > When running on low-power computing environments, it is often desirable to offload the computation to a third-party cloud environment to get the answer faster and use fewer local resources. However, this may be problematic if one does not fully trust the available cloud environments. Homomorphic encryption (HE) is the ideal means to alleviate this problem, providing cryptography for computations. HE is currently more expensive to perform than running a neural network itself, defeating its own utility in this scenario. CSPS provides a heuristic means of obscuring the nature of the input (content), and output (number of classes/prediction),  while also reducing the total local compute required.
>
> **4.2.2:**
> >Extreme Multi-label  (XML) is the scenario where, given a single input of size $d$, $C >> d$ classes are used to predict. This is common in e-commerce applications where new products need to be tagged, and an input on the order of $d \approx 5000$ is relatively small compared to $C \geq $ 100,000 or more classes. This imposes unique computational constraints due to the output space being larger than the input space and is generally only solvable because the output space is sparse --- often less than 100 relevant classes will be positive for any one input. VSAs have been applied to XML by exploiting the low positive class occurrence rate to represent the problem symbolically.

---

### Official Review · Reviewer_rBtB · 2024-06-26

**Soundness:** 3
**Presentation:** 2
**Contribution:** 3
**Rating:** 8
**Confidence:** 5

**Summary:**

This work describes a novel vector-symbolic architecture with linear-complexity binding/unbinding operations which includes the following components:
- A novel binding/unbinding scheme based on the Walsh-Hadamard transform, which reduces to element-wise multiplication and division due to the self-inverse properties of the Hadamard matrix, the definition of the binding operation, and the proposed projection step.
- Initialization of hypervector entries following a bimodal Gaussian distribution for purposes of numerical stability.
- A hypervector projection step which in expectation reduces the amount of noise which is incurred by unbinding a vector from a sum of bindings of pairs of vectors. It also simplifies the final form of the bind/unbind operations.
- A correction factor that is applied to augment the similarity score when retrieving from a sum of bound vectors, when the number of terms in the sum is known.

The authors experimentally test the effectiveness of their novel VSA scheme in several tasks including pure synthetic VSA tasks, as well as two tasks in which VSA schemes have previously been combined with deep neural networks.

**Strengths:**

- The paper is written clearly.
- The experimental results in section 4.2 are convincing.
- The noise-reducing projection step is an interesting idea. Although the idea is not novel in itself, the definition of the specific projection step is novel as it is well applied in this novel context of Hadamard-derived binding.
- This VSA is highly efficient, including only element-wise operations.
- The final HLB bind/unbind operations are trivially simple, but there is a novelty and potentially useful insight in their derivation.

**Weaknesses:**

1) Perhaps, it is worth mentioning and if possible elaborating/comparing with other parameterizable binding schemes such as "Steinberg and Sompolinsky, Associative memory of structured knowledge. Sci Rep 12, 21808 (2022)". This binding scheme is based on a trainable matrix. In fact, this parameterized binding scheme is generic by providing a continuum choice from d (VSA) to d^2 (TPR) for d-dimensional embeddings.

2) Besides the synthetic VSA tasks, out of the studied VSA+deep-learning applications, the XML classification results look very promising and practical (although CSPS looks interesting, it is a pseudo-encryption and hence heuristic). The reviewer was wondering if the proposed HLB binding and unbinding operators could have an edge in other hybrid VSA+deep-learning architectures such as:
    - [27] where MIMOConv used HRR-based binding and MBAT-based unbinding, and MIMOFormer used MAP-based binding and unbinding
    - Another neuro-symbolic capability of HRRs was studied in "M. Hersche, et al. A neuro-vector-symbolic architecture for solving Raven’s progressive matrices. Nat Mach Intell 5, 363–375 (2023)". Particularly, fractional power encoding was used to describe a set of arithmetic rules with binding and unbinding for abductive reasoning.
    - Another closely related work to XML classification in which the number of classes C is larger than dimension (d << C) is "M. Hersche, et al. Factorizers for Distributed Sparse Block Codes. Neurosymbolic Artificial Intelligence, 2024". Would be interesting to see how HLB can handle sparse (block-wise) codes.

3) The other key issue with the paper is flaws in its presentation. Some of them are:
    - The sentences starting at lines 39 and 40 are fully out of place.
    - Line 107 mentions an “unburdening” operation, clearly referring to “unbinding”.
    - Line 197 "HLBuses" needs spacing
    - In Theorem 3.1, the \cdot multiplication notation was used in place of the \odot for element-wise multiplication used otherwise in the paper.
     - Properties 3.1 is formulated in a weird way: “Let x sampled from omega holds the following properties…”
     - In properties 3.1, it seems that an expectation operator is missing in the final property concerning the L2-norm of x sampled from MiND.

4) The blue horizontal strip at the top of Figure 4 is a weird anomaly which was not addressed by the authors.

**Questions:**

- It would be great to elaborate on other trainable biding schemes discussed in (1) weaknesses
- If time and compute resources allow, it would be interesting to have more results based on the VSA neuro-symbolic architectures pointed out in (2) weaknesses, especially since the first two works have code repositories publically available.
- What is the reason behind the blue horizontal strip at the top of Figure 4?
- I did not fully understand what the hyperparameter search procedure for experiments in Section 4.2 was. Did the authors re-use the hyperparameters from the experiments in the referenced papers?
- In which scenarios is the discussion in section 3.2 relevant? The proposed factor only scales the similarity values, it does not modify the rankings of similarity scores, so it is not entirely clear to me in which context this matters.

**Limitations:**

This work has no negative societal impact.

---

> ### Author Rebuttal · Authors · 2024-08-06
>
> **W1:** The noted work is indeed valuable, though we can not implement it's experiments in the short rebuttal time. We will add the following paragraph to the manuscript.
>
> > As noted in (Steinberg and Sompolinsky), most VSAs can be viewed as a linear operation where $\mathcal{B}(a,b) = a^\top G b$ and $\mathcal{B}^*(a,b) = a^\top F b$, where $G$ and $F$ are $d \times d$ matrices. Hypothetically, these matrices could be learned via gradient descent, but would not necessarily maintain the neuro-symbolic properties of VSAs without additional constraints. Still, the framework is useful as VTB, HRR, MAP, HLB, and the original TPR all fit within this generalized representation. By choosing $G$ and $F$ with specified structure, we can change the computational complexity from $\mathcal{O}(d^2)$ (like TPR) to log-linear (like HRR), or $\mathcal{O}(d)$ for HLB and MAP.
>
> **W2** We will add a discussion of all these relevant related works to the paper and the wide and growing interest and applicability of VSAs in deep learning.
>
> We have experiments for MAP-B in progress answering a question of reviewer xpT1. Our available computing resources are limited as we are trying to work through the MIMOConv code and make sure that we have altered it correctly and scheduled an experiment.
>
>
>
>
> **W3:** Thank you for the typo identifications, they have been corrected!
>
> **W4:** We were not able to determine why the non-projected Hadamard binding seems to work well for seemingly random and arbitrary dimension sizes of $d$ (not for a lack of trying, we spent about a month trying to understand what was happening without success). In all cases, our projected Hadamard ($\eta^\pi$ noise term) has lower error and behaves consistently across $\rho$ and $d$, and resolved the asperity we saw with the projection-less Hadamard. For this reason, our final HLB is superior to the intermediate method, so we did not consider it further. We will add this context to that appendix section.
>
> **Q1:** see W1
>
> **Q2:** see W2
>
> **Q3:** see W4
>
> **Q4:** Yes, we re-used the hyper-parameters from the prior papers. These were primarily the network architecture size and number of layers. All were trained with Adam and used the standard recommended learning rate.
>
> **Q5:** The rescaling is useful as it allows the designing of a larger system to easily reason about their architecture. For example, if the unscaled dot product went into a softmax or other non-linearity, the expected behavior would change when the number of neurons/dimension $d$ changed. Having the scaling factor means that one can always (subject to noise) expect near-zero/one values for missing/present inputs. This is also critical to avoid vanishing gradients in the backward pass of a differentiable system.

---

> > ### Comment · Reviewer_rBtB · 2024-08-11
> >
> > Thanks for the clarifications. We want to re-emphasize the importance of this work that reduces the computational complexity of binding and unbinding from longstanding log-linear (HRR) to linear (HLB). This reduction will surely have profound implications for the use of VSA, especially when integrated with large deep learning models where binding and unbinding could quickly be a computational bottleneck. Due to these broad impacts, I will increase my score to Strong Accept.

---

> > > ### Author Response · Authors · 2024-08-12
> > > **Thank you for score raise and update!**
> > >
> > > Thank you for the score raise; we will indeed emphasize this in the paper and are glad we could answer your questions.
> > >
> > > We also have a small update: In running the original MIMO-nets code, the ETA for replicating their results is 191.5 hours. This is longer than the paper reported, so there may still be issues to resolve on our end or some missing detail. We will still endeavor to determine the discrepancy and explore MIMO-nets and other approaches.
> > >
> > > Thank you again for your review, and we hope to present this work at the conference!

---

### Official Review · Reviewer_6ktE · 2024-07-08

**Soundness:** 3
**Presentation:** 2
**Contribution:** 3
**Rating:** 6
**Confidence:** 3

**Summary:**

The authors propose a new form of Vector Symbolic Architecture (VSA), which leverages the Walsh Hadamard transform for vector binding. The new binding is named Hadamard-derived linear binding (HLB), and it achieves comparable or better performance than existing VSAs when performing classic VSA tasks and combined with deep learning.

**Strengths:**

The new binding method is developed with solid mathematical support. With the Hadamard-derived linear binding, the complexity of binding remains at O(d) while also providing some intriguing properties like the stability of similarity scores under the binding of multiple items. In the experiments, the new VSA is able to achieve better or comparable performance than existing VSAs such as MAP and HRR.

**Weaknesses:**

1. The motivation behind using WHT to derive the binding operation is missing.
2. Section 3 (the part before subsection 3.1) is rather hard to follow due to possible typos and poor organization.
3. It is unclear why the proposed method improves the performance in deep learning related tasks (in section 4.2).

**Questions:**

1. Is there any connection between the performance improvements in section 4.2 and the numerical stability of HLB?
2. Is there any benefit of having a symmetric binding?
3. In Definition 3.1, what does B=B* mean? Could you provide a more detailed proof of Theorem 3.1?
4. What are the differences between 'the proof of theorem 3.1' and equation (5)?

**Limitations:**

The limitations are not provided in the paper. The main limitation is that the merit of the new VSA is unclear in practice as the baselines only involve VSA.

---

> ### Author Rebuttal · Authors · 2024-08-06
>
> **W1:** We will add the below explanation motivating the choice of the WHT to the manuscript:
>
> > Our motivation for using the WHT comes from its parallels to the FFT used to derive the HRR and the HRR's relatively high performance. The Hadamard matrix has a simple recursive structure, making analysis tractable, and its transpose is its own inverse, which simplifies the design of the inverse function $\mathcal{B}^*$. Like the FFT, WHT can be computed in log-linear time, though in our case, the derivation results in linear complexity as an added benefit. The WHT is already associative and distributive, making less work to obtain the desired properties.  Finally, the WHT involves only $\{-1,1\}$ values, avoiding numerical instability that can occur with the HRR/FFT. This work shows that these motivations are well founded, as they result in a binding with comparable or improved performance in our testing.
>
> **W2:** We have corrected all noted typos and done another editing pass to make sure none remain.
>
> For the flow of Section 3, we will add the following content into the intro/starting paragraphs to help ease the reader through the section.
>
> >First, we will briefly review the definition of the Hadamard matrix $H$ and its important properties (see W1 above) that make it a strong candidate from which to derive a VSA.  With these properties established, we will begin by deriving a VSA where binding and unbinding are the same operation in the same manner as which the original HRR can be derived[30]. Any VSA must introduce noise when vectors are bound together, and we will derive the form of the noise term as $\eta^\circ$. Unsatisfied with the magnitude of this term, we then define a projection step for the Hadamard matrix in a similar spirit to [8]'s complex-unit magnitude projection to support the HRR and derive an improved operation with a new and smaller noise term $\eta^\pi$. This will give us the HLB bind/unbind steps as noted in Table 1.
>
> **W3:** We believe HLB performs better in deep learning tasks because it avoids exploding/vanishing gradients. This can be observed in Figure 3 where HLB has a consistent magnitude norm under multiple operations, and the avoidance of numerical instability with the FFT or other operations. We will add this note to the paper.
>
> **Q1:** We believe so; see W3 note. In particular, the low performance of MAP-C despite similar mechanical operation points to a primary difference being the scale-sensitivity based on the number of bound/bundled terms.
>
> **Q2:** There are pros/cons to symmetry in binding, as noted in [11]. Non-symmetric VSAs have an apparent advantage in representing hierarchical structures (e.g., a stack) with lower noise. However, they may not be as easy to use in certain symbolic contexts due to the lack of symmetry and the inability to unbind multiple values in a single operation. This will be added to the manuscript, though we note that we make no claim on a preference for one vs the other. We pursued a symmetric VSA in our work simply because we thought we could build one effectively.
>
> **Q3:** $\mathcal{B}=\mathcal{B}^*$ was stating that in the context we used the same function for both binding and unbinding. This will be clarified in the revision.
>
> This is the detailed step by step breakdown of the Theorem 3.1
>
> $\mathcal{B}^{*}(\mathcal{B}(x\_1, y\_1) + \cdots + \mathcal{B}(x\_\rho, y\_\rho), y\_i^\dagger)$
>
> $= \mathcal{B}^{*}(\frac{1}{d} \cdot H (H x\_1 \odot H y\_1)  + \cdots + \frac{1}{d} \cdot H (H x\_\rho \odot H y\_\rho), y\_i^\dagger) $
>
> $= \frac{1}{d} \cdot H ((H x\_1 \odot H y\_1 + \cdots + H x\_\rho \odot H y\_\rho) \odot \frac{1}{H y\_i})$
>
> $= \frac{1}{d} \cdot H (H x\_1 \odot H y\_1 \odot \frac{1}{H y\_i} + \cdots + H x\_i \odot H y\_i \odot \frac{1}{H y\_i} + \cdots + H x\_\rho \odot H y\_\rho \odot \frac{1}{H y\_i}) $
>
> $= \frac{1}{d} \cdot H (H x\_1 \odot H y\_1 \odot \frac{1}{H y\_i} + \cdots + H x\_i + \cdots + H x\_\rho \odot H y\_\rho \odot \frac{1}{H y\_i})$
>
> $= \frac{1}{d} \cdot H (H x\_i + \frac{1}{H y\_i} \odot \sum\_{\substack{j=1,j \neq i}}^{\rho} (H x\_j \odot H y\_j))$
>
> $= x\_i + \frac{1}{d} \cdot H (\frac{1}{H y\_i} \odot \sum\_{\substack{j=1, j \neq i}}^{\rho} (H x\_j \odot H y\_j) )$  Applying Lemma 3.1
>
> Then we get $= x\_i $ if $\rho = 1$, but otherwise we have the noise term for $\rho > 1 $ where $x\_i + \eta\_i^\circ $. Where $\eta\_i^\circ = \frac{1}{d} \cdot H (\frac{1}{H y\_i} \odot \sum\_{\substack{j=1, j \neq i}}^{\rho} (H x\_j \odot H y\_j) )$
>
> **Q4:** Equation (5) has the same operations as theorem 3.1, except Equation (5) includes a projection step when theorem 3.1 does not. It shows that we end up with a different description of the error term $\eta$ when adding a projection step that is easier to analyze and work with. This will be added to the revision.

---

> > ### Comment · Reviewer_6ktE · 2024-08-08
> >
> > I am satisfied with the response from the authors, which answers all my questions. The further clarification of the motivation, mathematical derivation, and experiments have made this paper even stronger. Therefore, I changed the score accordingly.

---

> > > ### Author Response · Authors · 2024-08-10
> > > **Thank you!**
> > >
> > > We are glad we have satisfied your questions and are very appreciative of the score raise! Please let us know if there is anything else that comes up.

---

### Official Review · Reviewer_xpT1 · 2024-07-09

**Soundness:** 3
**Presentation:** 3
**Contribution:** 3
**Rating:** 7
**Confidence:** 4

**Summary:**

The paper proposed HLB, a vector symbolic architecture derived from the Hadamard transform, reminiscent of holographic reduced representation, to mitigate the challenges that classical VSAs face in deep learning tasks such as numerical stability. Results show comparable memorization capability from alternative models and improved performance on two practical tasks, CSPS and XML.

**Strengths:**

1.	The paper is well-presented and delineates the design principles of HLB in the context of other VSA methods, making the position of the HLB clear.
2.	HLB is original in the sense that WH transform has not been leveraged directly to derive a VSA before (to the best of my knowledge); although the main design change from HRR is that the Fourier transform is replaced with the Hadamard transform, subsequent derivations of HDC properties and additional technique (the correction term, and the choice of sampling distribution of the hypervectors) is novel.
3.	The paper presents reasonably rigorous (see weakness) proof for its claim and supports it with empirical results. Experiments cover fundamental analytical properties desired from a VSA model as well as some applications of VSA in DL.
4.	The paper is significant in the field of vector symbolic architecture as a novel VSA model that may address some of the existing challenges in VSA, particularly the numerical stability problem.

**Weaknesses:**

1.	Several typos in the draft (line 87 it’s, 95 four, 150 dervise, 197 HLBuses, 260)
2.	Although the key parameter selection is explained in the experimental section, the authors defer the main experimental details to the referenced work. The authors are encouraged to briefly summarize each experiment and how HRR is used.
3.	See questions for additional concerns.

**Questions:**

1.	Is there a way to compute or estimate $\rho$ (number of bound items) given the vector? Many VSA can estimate via its norm and subsequent processing does not need to actively track the number of bound items in practice.
2.	How does HLB compare to other VSAs in computational complexity?
3.	Is MAP or MAP-C used in the experiment? They seems to be used interchangeably, but the C (which I assume to stand for “continuous” base vectors as opposed to bipolar) in MAP-C was never explained.
4.	Although the VSAs in comparisons are selected well, all three approaches leverage approximate unbinding while HLB leverages exact unbinding. How would HLB compare with alternative approaches in general, especially MAP (with bipolar vector initialization) and FHRR (HRR in the Fourier domain)? Both support exact unbinding (due to vectors having exact inverses).

**Limitations:**

No limitation section is present.

---

> ### Author Rebuttal · Authors · 2024-08-06
>
> **W1:** Thank you for the typo catches, they have been fixed!
>
> **W2:** For CSPS each network has four (convolutional) U-Net rounds in every
> experiment and doubles from 64 filters after each round,
> halving in size for the decode. The local prediction network has 4 convolutional layers, with max-pooling after each round, followed by three linear layers of 2048, then 1024, and finally, the $K$ classes of neurons.
>
> For XML, two hidden layers on a fully connected input for each dataset are used. The dimension $d=400$ for the smaller datasets and $d=3000$ for  Amazon-13K and larger datasets. This is significantly less memory than a $d \times L$ matrix for normal BCE where $L$ is up to 200k in our experiments.
>
> Each of 4.2.1 and 4.2.2 included this briefly in text, but was clearly not sufficient. For 4.2.1, we will add a new figure to help summarize (see one-page PDF). For 4.2.2., we will add a brief explanation, please see the one-page PDF again as the equations don't render properly in markdown/OpenReview.
>
> **Q1:** The 2-norm of the composite representation $\chi$ can be approximated as $\sqrt{d \cdot \rho}$.  Thus solving $\|\chi\|\_2 = \sqrt{d \cdot \rho}$ gives $\rho = \|\chi\|\_2^2 \cdot d^{-1} $. This is a good estimate of $\rho$ with a R-square value of $0.9865$, please see the rebuttal 1-page PDF for a plot of this relationship.
>
> **Q2:** HLB and MAP have $\mathcal{O}(d)$ complexity, whereas HRR is $\mathcal{O}(d \log d)$ (for an FFT), and VTB has $\mathcal{O}(d \sqrt{d})$ (for $\sqrt{d}$ matrix-vector products of $\sqrt{d}^2$ cost each). We will add this to the manuscript.
>
> **Q3:** MAP-C is used because it is the version of MAP that allows continuous vector values. Other variants of MAP require integer values, and thus can't be differentiated. So MAP-C is the logical point of comparison and is shortened to "MAP" in the manuscript. We will explain this in the revision; thank you.
>
> **Q4:**  For FHHR, we clarify that our experiments use the projected HRR of [8], which enforces complex unit magnitude -- and is thus equivalent to FHHR and uses an exact unbinding operation. Given that [8] and [1] both found the projection step necessary for good performance, we did not see value in running the original unprotected HRR.
>
> For MAP-B (MAP with binary initialization), we were able to run the CSPS experiment, which still shows MAP-B performing worse than our HLB. The result is below, and we are still running the XML experiments during the rebuttal phase. Once completed they will go into the main paper. Still, it is clear that MAP-B does not outperform MAP-C, let along our HLB, indicating the initialization was not a singularly important difference.
>
> MNIST: 98.40\%
>
> SVHN: 92.43\%
>
> CIFAR-10: 82.83\%
>
> CIFAR-100: 57.76\%, 84.63%
>
> MiniImageNet: 57.91\%, 82.81%
>
> GM: 75.90\%, 83.72%
>
> The XML results thus far:
>
> | Dataset | nDCG | PSnDCG |
> |---|---|---|
> | Bibtex | 59.412 | 46.340 |
> | Delicious | 65.431 | 32.122 |
> | Mediamill | 86.886 | 66.562 |
> | EURLex-4K | 71.128 | 26.340 |
> | EURLex-4.3K | 85.023 | 38.820 |

---

> > ### Author Response · Authors · 2024-08-12
> > **Small update, XML with MAP-B**
> >
> > We wanted to update you that the largest XML test compiled with consistent results: MAP-B Delicious-200K: nDCG 44.296, and PSnDCG  6.720, both below the scores of our HLB and not discernably worse/better than MAP-C. This would further support that the initialization difference was not a singularly key-factor in our improved performance.

---

> > > ### Comment · Reviewer_xpT1 · 2024-08-13
> > >
> > > Thank you for the response. The authors have answered all my questions and provided results that make the paper stronger. I have increased the score accordingly.

---

### Author Rebuttal · Authors · 2024-08-06

We are pleased all reviewers are interested in the paper and found it novel and significant in its results. All feedback was valuable and has been incorporated into a revised manuscript, with responses inline to each reviewer's individual questions. xpT1 please note your answers can be found in the one-page PDF due to formatting limitations of OpenReview.

In summary, reviewers identified some typos and sections where additional wording/content would help the reader understand the work better and follow the paper without having to refer to additional resources. Assuming the historical extra camera-ready page for NeurIPS, this space was used to add such additional text.

Two reviewers noted that other deep learning+ VSA architectures have been proposed, which could benefit from our VSA being applied to them. We are attempting to test some of these but coding/compute time is limited in the rebuttal window. All of these noted works will be added to the manuscript. In addition, additional results have been included in the rebuttal on MAP with binary initialization that shows our method's improvement over the MAP VSA is due to more than just the binary initialization.

---

### Decision · Program_Chairs · 2024-09-25

**Decision:**

Accept (poster)

**Comment:**

## Summary of the paper
This paper proposes a novel vector symbolic architecture (VSA) called Hadamard-derived Linear Binding (HLB) inspired by previous works on tensor product representations (TPR) and addresses some of their challenges with the deep learning architectures. The paper leverages the Walsh Hadamard transform for vector binding to learn vector symbolic representations. The paper empirically shows superior results on the typical VSA benchmark datasets and tasks.

## Verdict

Overall, the paper is well-written, and the approach proposed is novel and relevant to the broader AI/ML community. It touches upon an interesting problem of learning better VSAs. The results are convincing and show significant improvements over the previous TPR approaches. This work reduces the computational complexity of binding and unbinding from log-linear (HRR) to linear (HLB), which can be impactful. The reviewers had some concerns and questions, but the authors did a very good job addressing them during the rebuttal. Thus, I recommend this paper for acceptance.

There are some fixes I would recommend the authors to do for the camera-ready version of the paper:
1. Further clarification of the motivation, mathematical derivation, and experiments have strengthened this paper as pointed out by the reviewers.
2. (Minor) There are some typos in the paper, please fix them, for example, the sentence at the end of line 61 seems unfinished.